# Preparation of Beebread Caviar from Buckwheat Honey through Immobilization with Sodium Alginate

**DOI:** 10.3390/molecules25194483

**Published:** 2020-09-29

**Authors:** Małgorzata Smuga-Kogut, Agnieszka Pabiszczak, Maria Dymkowska-Malesa, Daria Szymanowska, Joanna Kobus-Cisowska, Judyta Cielecka-Piontek

**Affiliations:** 1Department of Agrobiotechnology, Faculty of Mechanical Engineering, Koszalin University of Technology, Raclawicka 15-17, 75-620 Koszalin, Poland; agnieszka040393@gmail.com; 2Department of Dietetics, The Institute of Sport and Health Studies, State University of Applied Sciences in Koszalin, Leśna 1, 75-582 Poland, Koszalin; mdymkmalesa@tlen.pl; 3Department of Biotechnology and Food Microbiology, Poznan University of Life Sciences, Wojska Polskiego 48, 60-627 Poznan, Poland; daria.szymanowska@up.poznan.pl; 4Department of Gastronomy Sciences and Functional Foods, Poznan University of Life Sciences, Wojska Polskiego, 60-637 Poznan, Poland; joanna.kobus-cisowska@up.poznan.pl; 5Department of Pharmacognosy, Poznan University of Medical Sciences, Swiecickiego 4, 60-781 Poznań, Poland; jpiontek@ump.edu.pl

**Keywords:** beebread caviar, buckwheat honey, immobilization, functional food

## Abstract

Honeys have a pleasant taste and a wide range of use. They are characterized by a relatively high consumption compared to bee pollen or beebread. Honeys are the most popular bee products. Considering health reasons, beebread exhibits the strongest properties as it has the highest nutritional value as well as strong detoxifying, antioxidant, and antiradical properties. Despite having such valuable properties, consumption of beebread is negligible; sometimes, it is limited only to supplementation in case of diseases. This paper proposes a new food product, that is, beebread caviar made from buckwheat honey. The expiry date and sensory and physicochemical quality of beebread caviar have been determined in this study. Beebread caviar, obtained by immobilization on alginate carrier, contained 0.34 mg GAE/mL extract. It remained stable until five days after preparation. Its total acidity was 33.7 mval/kg. Its extract content was 22.53%. Caviar had a high overall sensory score of 4.8 points on a 5-point scale. Beebread caviar can be successfully classified as probiotic food because beebread contains a large amount of lactic acid. In the form of caviar, a new, attractive, and convenient form of beebread consumption could become one of the products of comfortable and functional food.

## 1. Introduction

Bee products are rich in vitamins, minerals, and many bioactive substances. Honey, beeswax, and propolis have been used for thousands of years; however, the unique therapeutic and dietary properties of other less known bee products, such as royal jelly, hive products, and beebread, have been recognized only recently. Beebread is a mixture of pollen, honey, and throat secretions of bees, fermented in the hive, which makes it a rich source of proteins, sugars, lactic acid, vitamins, macro- and microelements, enzymes, and phenolic compounds. Such natural fermentation of beebread has a great influence on the assimilability of its nutritional values and activity of bioactive substances [1,2]. Thus, it has the most beneficial health properties among all bee products. It can be successfully used to treat many diseases and ailments. Despite its valuable properties, beebread is not a common dietary supplement, and its consumption is less when compared to popular honey [3]. The beebread sold in the market is in the form of brown, dry, and hard granules having an intense floral smell and an unpleasant taste. Preparing beebread for consumption involves grinding, soaking for several hours, and combining with other basic products (e.g., water, juice, and milk) to make the taste of the beebread more acceptable. In addition, during the preparation, special attention should be paid to the temperature, which must not exceed 50 ℃, as excessive heating of beebread can lead to a significant loss of its nutritional value. Such a long and tedious process of beebread preparation may discourage its regular supplementation. The creation of an easy-to-use beebread product in the form of comfortable and functional food, having a pleasant taste and smell, could increase not only the popularity of this valuable bee product but also its consumption in the daily diet. In terms of antioxidant properties and assimilability of nutrients, beebread is ahead of bee honey and pollen [4]. Regular consumption of beebread can be one of the methods for supplementing the daily diet with antioxidative compounds. Despite beebread’s excellent health properties, the scientific community is little interested in its study. Beebread composition varies according to the origin of the pollen but is mainly composed of water, proteins, carbohydrates, lipids, inorganic elements, and various other minor components such as decanoic acid, gamma globulin, nucleic acids, vitamins B and C, pantothenic acid, biopterin, neopterin, acetylcholine, and reproductive hormones, among others [5,6]. Considerable amount of information can be found in the literature on the types of polyphenol compounds contained in bee bread [7,8,9]. The most important of these compounds are: p-coumaric acid (367 µg/g), kaempferol (492 µg/g), isoramnetin (1086 µg/g) among phenolic compounds, as well as ferulic acid, caffeic acid, apigenin, and quercetin present in trace amounts, were identified in the composition of beebread. It is highly probable that some parts of the polyphenols contained in beebread could not be detected as they may be found in more complex substances, such as glycosides. Such solutions are widespread among plants [3].

Immobilization, also called spherification in the molecular cuisine, is an excellent method of making food products more attractive while keeping their all nutritional values intact [10,11]. The mixture of a given product with a carrier in appropriate proportions is condensed into a solution of sodium chloride in order to cross-link it. As a result, small, velvety gel balls, with a liquid interior of any taste, are formed. This is called caviar. This method is widely used in the molecular cuisine to produce caviar in various flavors, and in biotechnology, to immobilize enzymes or microorganisms to increase their activity [12]. Alginate is widely used in various industries such as food, beverage, textile, printing, and pharmaceutical as a thickening agent, stabilizer, emulsifier, chelating agent, encapsulation, swelling, a suspending agent, or used to form gels, films, and membranes [13,14]. Sodium alginate is the most common salt from alginate [15]. The U.S. Food and Drug Administration (FDA) classifies food grade sodium alginate as a GRAS (generally regarded as safe) substance in Title 21 of the Code for Federal Regulations (CFR) and lists its usage as an emulsifier, stabilizer, thickener, and gelling agent. The European Commission (EC) listed alginic acid and its salts (E400–E404) as an authorized food additive [16]. Immobilization of the mixture of beebread and honey on an alginate carrier is proposed to create an innovative product with a functional character [17]. Immobilized inside the caviar, nutrients and bioactive substances contained in the beebread retain their properties, and the taste of unattractive beebread becomes pleasant and interesting. Caviar from beebread facilitates the latter’s daily use in order to overcome the deficiency of important nutrients and prevent many diseases. Moreover, the product prepared in this way can be successfully stored for a long time, without worrying about the loss of valuable properties contained in beebread.

## 2. Results and Discussion

Immobilization is a technique where a mixture of different substances is coated inside another material. In the food industry, this method is frequently and willingly used. First, it solves problems resulting from limited chemical and physical stability of active food ingredients and limited compatibility between the active ingredient and the food substrate. Second, immobilization controls the release of sensory active substances as well as the bioavailability of nutrients [18]. Buckwheat honey contained 18.1% water, 52.7 mg/100 g proline, glucose content was 33.5 g/100 g honey, fructose content was 36.7 g/100 g, and sucrose below 0.5 g/100 g. The diastase number for buckwheat honey was 64.5 Schade units. In this study, caviar was created from beebread, the main base of which was a mixture of beebread and buckwheat honey (Figure 1). It consisted of 17% of beebread and 83% of honey. The proportions of the two components were kept thus in order to maintain a semi-liquid consistency of the material that allows spheres to form during the application of the mixture in calcium chloride solution. On the one hand, too much beebread addition made the mixture too thick, which made the immobilization process impossible. On the other hand, a large amount of honey was important for the sensory and physicochemical values of the resulting caviar. The intense sweetness and delicate, characteristic bitterness of buckwheat honey effectively masked the astringent taste from propolis, that is, the taste of beebread, making the whole a much more acceptable taste composition.

The concentration of sodium alginate used in the immobilization process depended on the substance that is subjected to this process. Too low concentration of sodium alginate solution may cause the caviar balls to break down when they are dropped into a calcium chloride bath. Similarly, too high concentration of alginate can cause too strict a consistency of caviar and result in absence of the feeling of its cracking in the mouth. In molecular gastronomy, 1% calcium alginate solution is most commonly used for the spherification of juices and other homogeneous products [19]. However, in the case of a thick mixture of beebread and honey, this concentration was insufficient, as the caviar balls decayed in a calcium chloride bath. Therefore, a 2% solution of sodium alginate was used, which is also used in industrial biotechnology, for example, to immobilize microorganisms [20]. 

In the form of a solution of sodium alginate, the carrier is to take part only in the structure-forming process of caviar balls. In any case, the predominant component cannot be palpable in the taste of the resulting beebread caviar [21]. Three attempts of spherification in different proportions of the base to the carrier were made, that is, 1:4, 1:2, and 1:1. The sample with proportions of 1:4 was characterized by the brightest color and the least intense taste; moreover, too high alginate content caused the disappearance of a liquid interior in the caviar ball. The consistency was uniform and the effect of cracking in the mouth did not occur. Despite the fact that the amount of carrier was halved in the next sample, caviar had a very similar consistency to caviar prepared in a ratio of 1:4; however, the taste and color were much more intense. The most favorable proportions of sodium alginate solution to the mixture of beebread and honey were 1:1. Caviar formed in the last sample (1:1 proportion) was characterized by a delicate consistency with a thin membrane surrounding the liquid interior of the ball, which cracked in the mouth. Both the taste and color of caviar were intense, characterizing beebread combined with buckwheat honey. The final result of all three samples is shown in Figure 2.

Regardless of the applied ratio of the taste base to the carrier, each time during caviar production, the problem of dosing the mixture to calcium chloride solution was faced. The dense and heterogeneous consistency of ground beebread combined with honey made it difficult to drop the mixture so that it obtained quite round shapes of caviar balls. Konik [22] created caviar from orange juice using a classic caviar box. The product obtained in this way was characterized by an ideal round shape, and the balls were small and even. In the case of the production of beebread caviar, the use of this method did not work and the use of dropping with a syringe, with a piston, and a much wider dosing hole, was used. Finally, slightly elongated and sometimes even tear-shaped caviar was obtained, which did not affect the sensory impressions accompanying the consumption of the product. The shape of the thus-obtained caviar was basically to its advantage and looked very interesting, as it was completely different from the traditional caviar obtained by the spherification method. 

During the microscopic examination of beebread caviar, the distribution of pollen grains, their size, and the thickness of their envelopes were compared. The examination was performed shortly after the preparation of caviar and again after storing it for three days. Moreover, the microscopic images of the entire caviar balls were analyzed (Figure 3 and Figure 4). 

The microscopic analysis showed that pollen grains were evenly distributed over the entire area of caviar balls. This proves that the same nutritional value is provided every time caviar is consumed. Visible pollen envelopes are thin and can be distinguished from the whole. After three days of storage, it can be observed that the color of the caviar changed. It became more amber, which proves that not only pollen, but also honey was hydrated [3,23]. A slight difference can be observed in the thickness of the pollen envelopes, and in their even distribution. The beebread, just after the preparation, was visible in the form of clusters and larger agglomerates, which were broken up three days after preparation. The pollen envelopes became slightly thinner. Grains were larger and more swollen because the interior of the caviar is liquid and allows the pollen grains to get wet. This is very important for assimilability of nutrients and bioactive substances. Immobilization of beebread in an alginate capsule saves time for its preparation for consumption, and storing it for a longer time positively affects the antioxidant properties of the product. 

The study of the content of phenolic compounds in beebread caviar showed an increase in the content of these compounds with the storage time. The lowest content of phenolic compounds was noted in caviar after preparation and it amounted to 0.34 mg GAE/mL extract. After five days of storage, the content of phenols increased almost threefold and amounted to 0.94 mg GAE/mL extract, which indicates a significant increase in the activity of bioactive compounds. After another five days of storage, the content of phenolic compounds decreased to 0.64 mg/mL, but still remained almost twice as high as the initial value (Figure 5). 

Bee honey has a high content of polyphenols and strong antioxidant properties. The antioxidant activity of honey in most sources is between 10 and 45%, and depends mainly on the type and area of origin of the honey studied [24,25]. According to the study by Socha et al. [26], the content of phenolic compounds in multiflower honeys is 47.13 mg GAE/100 g, on average. Similar results were obtained by Kieliszek et al., Majewska et al., and Wilczyńska [27,28,29] In this study, buckwheat honey was used, which, depending on the place and time of harvest, has a similar content of phenolic compounds compared to multiflower honeys. Beretta et al. [30] showed that the content of phenols in buckwheat honey was at a level of 48.22 mg GAE/100 g; whereas, according to Zujko et al. [31], the average phenols content was 95 mg GAE/kg. An effective method of enriching the honeys with antioxidant compounds is the addition of beebread, as the latter has the highest antioxidant capacity and the highest content of phenolic compounds among all bee products available in the market [26].

According to Majewska et al. [28], the highest antioxidant activity (91%) among the available bee products was observed in the case of beebread dissolved in honey. Similar results were obtained by [32]. Averaging the result for three samples of beebread from different regions of Lithuania, they obtained the value of antioxidant activity at the level of 93%. Socha et al. [26] also conducted a study on honey enrichment with beebread, in which an increase in the total content of phenolic compounds can be clearly observed. The average content of phenols in beebread-enriched honeys was 109.07 mg GAE/100 g; whereas, according to Ivanišová et al. [33], the content of these compounds in the beebread itself is between 12.4 and 25.4 mg GAE/g. Socha et al. [26] also claim that enriching honey with beebread is the most natural way to use the potential of beebread to supplement the diet with a variety of biologically active compounds. The addition of beebread causes a clear increase in the total content of phenolic compounds, including phenolic acids and flavonoids, and increases antiradical, antioxidant, and reducing activity. 

In this study, it was shown that the content of phenolic compounds in beebread caviar and buckwheat honey depends not only on the type and origin of raw materials used but also on the time of storage. An increase in the content of phenolic compounds in caviar was observed after five days of storage; whereas after 10 days, the product did not maintain antioxidant stability. Therefore, due to the bioactive ingredients of beebread, it is best to consume it within more than 24 h after immobilization and up to 5 days, when stored in refrigerated conditions at 7 °C. According to Bonin [20], the immobilization of bioactive substances allows the prolonging of their activity and stability, so that they can be used more effectively for a longer period of time. Storing immobilized beebread has a positive effect on the nutritional value and antioxidant properties of caviar. Moreover, by prolonging the time of beebread staying in the liquid environment, the assimilability of polyphenolic compounds and vitamins and minerals contained in it is increased. 

The content of extract and free acidity in beebread caviar were examined in three storage periods. Shortly after the preparation, acidity of caviar was 33.7 mval/kg, which increased as the storage time increased. For bee products, the value of free acidity should not exceed 50 mval/kg [34,35]. After five days of storage, the total acidity of the product increased rapidly up to 68 mval/kg. Such a significant change indicates the fermentation process inside the caviar ball. Fermentation mainly affects honey that is dissolved in water. In addition, the beebread contains enzymes that can break down or biotransform the components of bee pollen into organic acids, which increases the acidity of the product. Although the acidity increases, the sensory aspect of the product does not deteriorate. After 10 days of storage, the acidity also increased, but not as much as in the first 5 days. The acidity value on the tenth day of storage was 70.3 mval/kg (Figure 6).

Beebread caviar can be successfully classified as probiotic food, as beebread contains a large amount of lactic acid. Microorganisms contained in caviar, through their activity, can have a beneficial effect on health through the digestive system by regulating the balance of intestinal microflora [36].

The extract content on the day of caviar production was highest and amounted up to 22.53%. It was 22.43% after five days. After 10 days of storage, the extract content increased significantly up to 20.73%, which means a decrease in sugar content and may indicate the beginning of fermentation of honey contained in the caviar (Figure 7).

While comparing the acidity results with the extract content, one can clearly observe the fermentation process that results in an increase in acidity and a decrease in the extract content. The ongoing fermentation process causes a decrease in the content of sugars in the product, and an increase in the content of free acids, including lactic acid. This relationship was used in their studies by Samborska et al., Kruszewski et al., and Smuga-Kogut et al. [37,38,39] in production of honey powders. Consistency is one of the most important characteristics of caviar. Caviar evaluated shortly after its preparation had the best and most desired consistency. It was thin and imperceptible on the tongue membrane surrounding the liquid interior. The balls were firm and cracked under pressure, and thus they were rated at 5 points. After five days of storage, the balls became less firm and their outer membrane was thicker. Nevertheless, caviar still had a liquid interior and cracked in the mouth under pressure. Thus, it was rated at 4.33 points. After 10 days of storage, the product already became thicker and cracking the coating became more difficult. Furthermore, its interior was semi-solid. Thus, the grade awarded for this storage period was 3.33 points. The gradual hardening of the caviar balls with the extension of the storage time is a result of the natural properties of sodium alginate, which increasingly binds water and inevitably solidifies the product over time [40]. The taste of the obtained caviar remained characteristic of beebread. However, thanks to the addition of honey in an appropriate proportion, it became much more acceptable. It was intense, characterizing both the raw materials. The most perceptible flavor was sweet with a delicate sour aftertaste. The sour aftertaste was derived from the lactic acid naturally present in beebread. The taste of beebread caviar was extremely attractive and desirable for this type of product just after its preparation. Thus, it was rated at 4.67 points. After five days of storage, the taste of caviar still remained characteristic for both beebread and honey, but was much more intense and aromatic. Therefore, it was rated at 5 points. After 10 days of storage, the taste of caviar deteriorated. It was less noticeable and slightly less sweet, which lead it to be rated at 3.67 points. 

The aroma of beebread caviar immediately after preparation was characterized by an intense floral characteristic of all bee products. It was very sweet and pleasant and did not show any foreign smells. The assessment of the smell immediately after the preparation of the caviar was rated at 5 points. After five days of storage, the aroma became less intense and more delicate, but it was still characteristic of beebread and honey, which gave it a rating of 4.67 points. On the tenth day of storage, the caviar aroma was still pleasant and acceptable, but not so intense. It was delicately floral, but the smell of honey and beebread was difficult to distinguish, which gave it a rating of 3.67 points.

The product tested shortly after its preparation had the most attractive and desirable sensory characteristics, and thus, its overall rating was 4.73 points. At that time, caviar had a highly rated color, consistency, taste, and aroma. The fresh product not only retained the characteristics of ideal caviar obtained by the fermentation method but also the taste and smell of the raw materials used to make it, that is, beebread and honey. After five days of caviar storage, it was rated at 4.58 points, as the traits under study slightly changed. The color and taste showed a higher intensity and were evaluated higher. This happened due to appropriate soaking of pollen grains contained in beebread, which was the reason for an increased rating for these traits. The shape and smell were rated slightly lower, as they still had good sensory quality. The consistency of caviar deteriorated the fastest. It was no longer as attractive and characteristic of caviar as it was shortly after its preparation. After 10 days of storage, beebread caviar was characterized by the lowest sensory score, that is, 3.52 points, which means its lowest attractiveness for consumers. All the tested features were rated much lower, but were still positive and beneficial for the product. The product was still acceptable to consumers. The overall result of the sensory analysis of caviar is presented in Figure 8.

On the basis of the obtained results, a clear correlation was observed between the sensory attractiveness of beebread caviar and the length of storage. The highest quality of the studied product was characterized by up to 5 days of storage, and the lowest quality after 10 days of storage. Taste and color had very good sensory quality for up to five days of storage; a similar relationship can be observed in the case of smell and shape. The consistency of caviar, which is not favorably affected by the length of storage, deteriorates the fastest. Despite a gradual decrease in sensory evaluation, the product is not subject to spoilage and undesirable changes. It is only less intense in taste and smell, and less visually attractive.

## 3. Materials and Methods

Buckwheat honey, harvested in July 2018 (Słonino Apiary, West Pomerania, Poland), was used in the study. Buckwheat honey is classified as nectar honey (*Fagopyrum* pollen content—54.2%; *Trifolium* type pollen—28.3%, other pollen—17.5%). It was characterized by its dark color and sharp taste, as well as by the intense aroma of buckwheat flowers. It had a thick, liquid consistency. 

Beebread used in this study came from the Jezyce apiary (Darłowo, West Pomerania, Poland). It consisted of 180 bee colonies. The beebread was collected at the turn of June and July 2018. It had the form of small, hard brown granules. Its taste and smell were intensely floral with a palpable aftertaste of honey. Beebread was collected from the hives located near a buckwheat plantation. Hence, the majority of flower pollen observed under microscope originated from buckwheat flowers (*Fagopyrum sagittatum*). Therefore, the beebread was dissolved in buckwheat honey. Sodium alginate (Agnex, Bialystok, Poland) was used for immobilization. Sodium alginate (E401) was an odorless and colorless substance from which a 2.5% aqueous solution was prepared. Caviar was formed in a 2% calcium chloride solution (E509) (Stanlab, Lublin, Poland).

### 3.1. Immobilization Process

To produce caviar on the basis of beebread and buckwheat honey, a 2% solution of sodium alginate was used, to which 20 g of ground beebread, previously mixed with 100 g of buckwheat honey, was added. The resulting mixture was dropped at a 2.5% calcium chloride solution. After dropping, the material was conditioned in calcium chloride solution for 15 min, which was counted from the last obtained ball. Then, the prepared medium was rinsed several times on a sieve with distilled water to get rid of the salty taste of calcium chloride. Caviar was packed in glass jars with 100 g volume with twist off cap and was stored in a refrigerator at a temperature of 7 °C, in darkness. The experiments on antioxidants were performed after 5 and 10 days of storage.

### 3.2. Analytical Procedures

Buckwheat honey was analyzed according to the official Polish methods [41] in order to determine moisture-water content (refractometric method), diastase activity and proline (colorimetric method). Proline determination was performed after its separation from other amino acids present in honey, with spectrophotometric method, utilizing a UV-VIS 1600 spectrophotometer (VWR International, Gdansk, Poland). The same spectrophotometer was utilized in cuvette tests of glucose, fructose and sucrose were obtained by the enzymatic determination method using a sucrose/d-glucose/d-fructose UV test no 716260 (Boehringer Mannheim, R-Biopharm AG, Darmstadt, Germany). Diastatic activity of honey was determined using spectrophotometry, in which insoluble starch conjugated with blue dye was used as the substrate. It was hydrolyzed by amylase, which leads to obtaining water-soluble fragments of starch chain, creating blue connections with the dye, and the absorbance of which was measured at wavelength of 620 nm. The solution absorbance is proportionate to the diastatic activity of the sample [9].

The beebread immobilized on alginate capsules was analyzed for the content of phenolic compounds by the Folin–Ciocalteu method (AOAC, 1974). Briefly, 10 g of sample was mixed with 20 mL of methanol and the mixture was stirred for 30 min at 30 °C. Then, 250 µL of supernatant, 250 µL of Folin–Ciocalteu reagent, and 500 µL of 20% sodium carbonate in water were added in 4.00 mL of water. After 30 min, absorbance was measured at 760 nm using UV–vis spectrophotometer with methanol as the reference. Gallic acid (0–100 mg/L) was used to produce a standard calibration curve. The total phenolic content (TPC) was expressed in milligrams of gallic acid equivalents (mg GAE/mL of extract). Total acidity was determined according to Polish Standard [9], according to which 10 g of caviar was weighed, disintegrated, and dissolved in 50 mL of distilled water, followed by 15 min shaking on a Vortex. Subsequently, the samples were titrated with 0.1 M sodium hydroxide against phenolphthalein as an indicator to bright pink coloration. The titration was performed in three replications. Total acidity was calculated following a formula, including the sample size and the amount of sodium hydroxide used to produce discoloration. The extract was determined using a HANNA HI 96803 digital refractometer (HANNA Instruments, Olsztyn, Poland). To make this possible, a sample was disintegrated to liquefy it and it was then placed using a plastic pipette on a dry and clean prism. The measurement result was automatically read and was performed at a calibration temperature of the device, that is 20 ℃. The measurement was done in three replications. Sensory analysis was performed with a five-point scale method [10], which includes five quality levels for each quality trait. Appropriate quality definition is assigned to each level. These definitions are specific for different types of products, thus they have to be strict and separate for the individual levels of the scale, so the assessing person should not have any doubts as to which level the product should be qualified. The point assessment was performed by a team of 5 people, and the results presented in the publication were averaged. All analyses were performed in three repetitions, that is, on the day of caviar preparation and after 5 and 10 days of storage. A microscopic analysis of caviar balls was performed in order to check uniformity of the distribution of beebread particles and estimate the ratio of solid-to-liquid fractions during storage in each of the caviar balls. Statistical analysis was performed in order to compare the content of phenolic compounds and extract, and determine the differences in acidity in the finished product at different storage times. The mean values of the given physicochemical parameters were compared and the significance of differences between the samples was demonstrated. The statistical analysis was performed using Excel software. The student’s *t*-test was used at α = 0.05. The difference was considered statistically significant when *p* ≤ α.

## 4. Conclusions

The use of immobilization with sodium alginate made it possible to obtain caviar from a mixture of beebread with buckwheat honey, which is proposed as a new convenient and functional food product. The content of total phenolic compounds in beebread caviar was the highest on the fifth day of product storage and amounted up to 0.94 mg GAE/mL extract. Caviar with the total acidity of 33.7 mval/kg and the extract content of 22.53% was obtained from beebread. The physicochemical quality of beebread caviar was the highest after five days of product storage, whereas the highest sensory quality of the product was noted immediately after its preparation. The color and taste of the obtained beebread caviar were the most attractive after five days of product storage. Aroma and shape deteriorated slightly with the time of storage, whereas consistency deteriorated significantly with the time of storage.

## Figures and Tables

**Figure 1 molecules-25-04483-f001:**
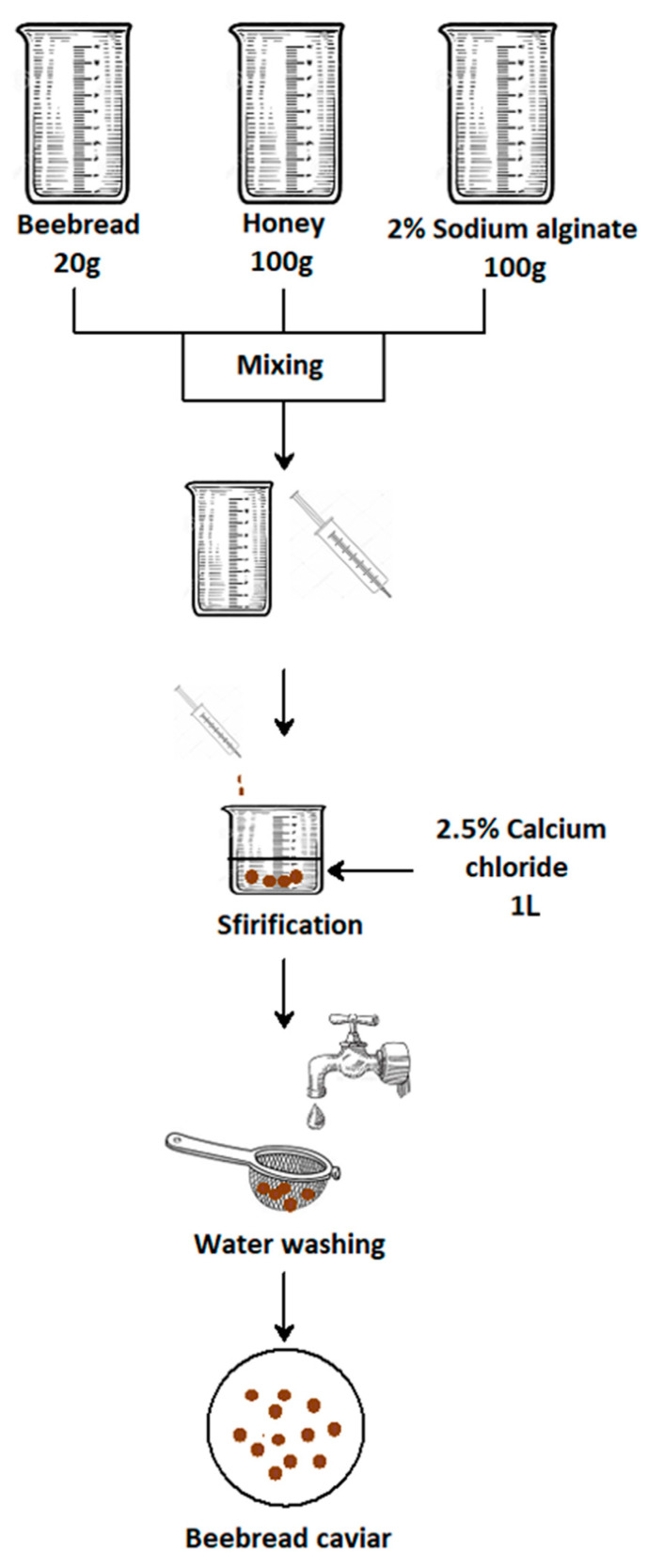
Schematic course of the process of beebread immobilization.

**Figure 2 molecules-25-04483-f002:**
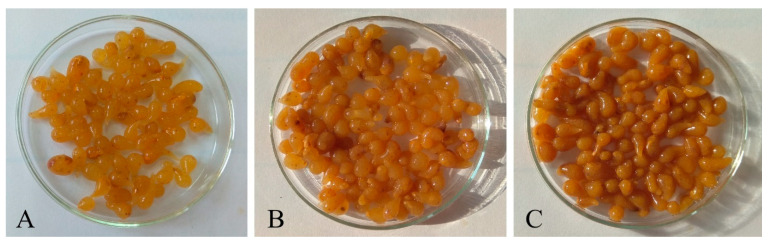
Beebread caviar in various proportions of the taste base to the carrier: (**A**)—1:4, (**B**)—1:2, (**C**)—1:1.

**Figure 3 molecules-25-04483-f003:**
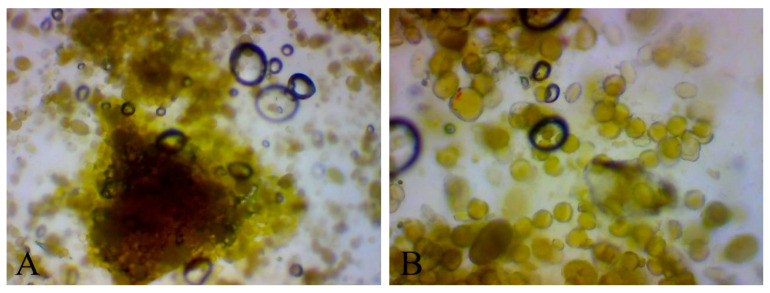
Microscopic images of beebread caviar just after preparation: (**A**)—4× magnification, (**B**)—10× magnification.

**Figure 4 molecules-25-04483-f004:**
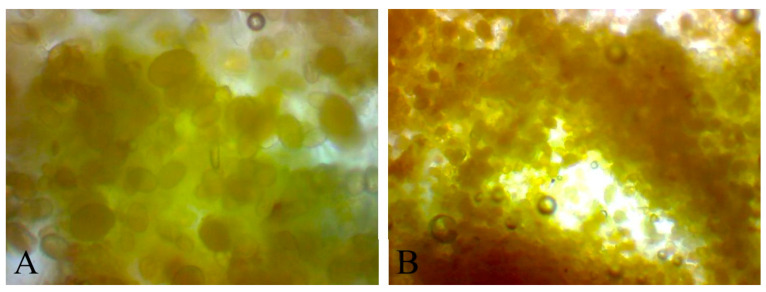
Microscopic images of beebread caviar after 3 days of storage: (**A**)—4× magnification, (**B**)—10× magnification.

**Figure 5 molecules-25-04483-f005:**
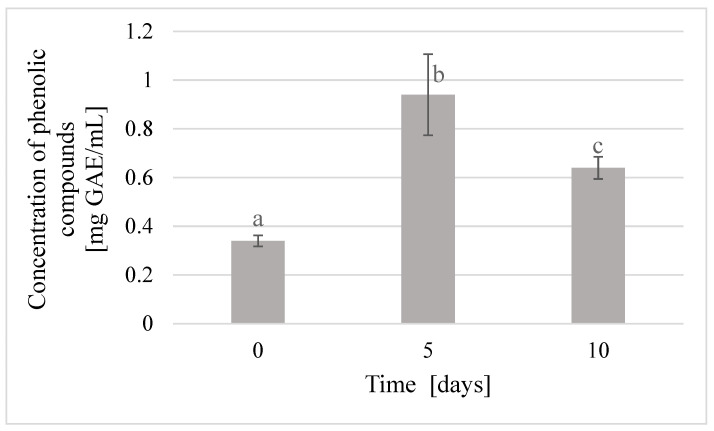
An effect of storage time on the content of phenolic compounds in beebread caviar; *p*_a-b_ = 0.00345; *p*_b-c_ = 0.04104.

**Figure 6 molecules-25-04483-f006:**
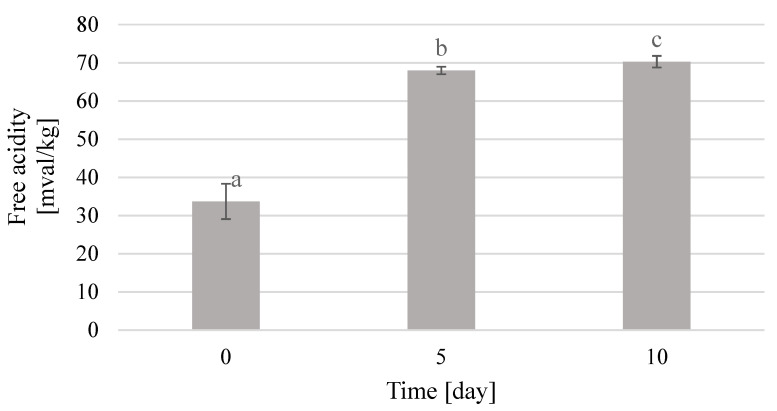
Changes in free acidity of beebread caviar during storage; *p*_a-b_ = 0.00023; *p*_b-c_ = 0.09126.

**Figure 7 molecules-25-04483-f007:**
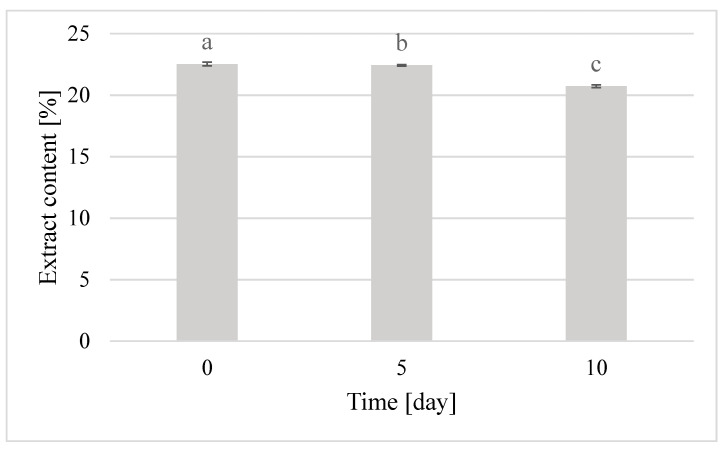
Changes in extract content in beebread caviar during storage; *p*_a-b_ = 0.34864; *p*_b-c_ = 0.00002.

**Figure 8 molecules-25-04483-f008:**
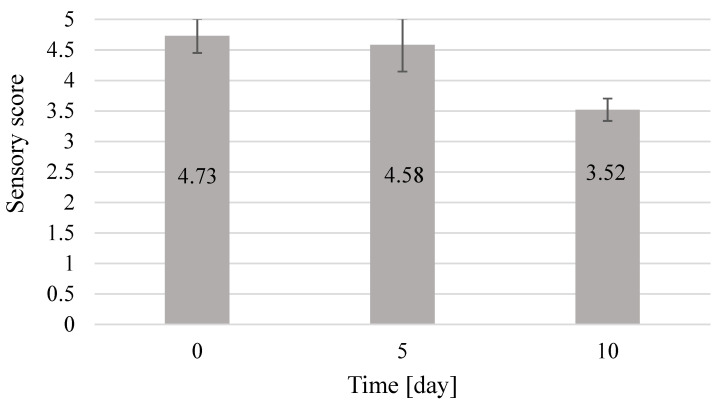
Overall result of the sensory analysis of beebread caviar.

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
