# Peer review of "Preparation of Beebread Caviar from Buckwheat Honey through Immobilization with Sodium Alginate"

_molecules, 2020, doi:10.3390/molecules25194483_

Round 1

Reviewer 1 Report

Comment to authors:

  1. The storage conditions during five and ten days should be mentioned.
  2. Line 56: “Beebread supplementation is the most beneficial way to make a polyphenols-rich diet.” rephrase.
  3. The acidity of the product increased with the storage time, is it favored for the consumers?
  4. Line 5-59: “Only two publications had been undertaken to study its composition through chromatography coupled with mass spectrometry”. In fact, several published papers presented on this subject as the following:
  • Othman, Z.A., Wan Ghazali, W.S., Noordin, L. and Mohamed, M., 2020. Phenolic compounds and the anti-atherogenic effect of bee bread in high-fat diet-induced obese rats. Antioxidants9(1), p.33.
  • Sobral, F., Calhelha, R.C., Barros, L., Dueñas, M., Tomás, A., Santos-Buelga, C., Vilas-Boas, M. and Ferreira, I.C., 2017. Flavonoid composition and antitumor activity of bee bread collected in northeast Portugal. Molecules22(2), p.248.
  • Hryniewicka, M., Karpinska, A., Kijewska, M., Turkowicz, M.J. and Karpinska, J., 2016. LC/MS/MS analysis of α‐tocopherol and coenzyme Q10 content in lyophilized royal jelly, beebread and drone homogenate. Journal of mass spectrometry51(11), pp.1023-1029.
  1. Using sodium alginate, and calcium chloride during process is safe?, explain
  2. Line 151 “also propolis was hydrated”, the relationship between propolis and caviar here is not clear, please clarify.
  3. The plant origin of the bee bread should be included.
  4. The content of phenolic compounds in beebread caviar increased with the storage time, explain.
  5. Is the final product expected to be safe, useful, cheap, and attractive of consumers?

  1. Figure1: The conditions of each step should be included and the figure must be done with high resolutions.
  2. References section’: styles should be done based on the journal requirements.
  3. The recent interesting reference may help in updating information in the introduction.
  •  Khalifa, S.A., Elashal, M., Kieliszek, M., Ghazala, N.E., Farag, M.A., Saeed, A., Xiao, J., Zou, X., Khatib, A., Göransson, U. and El-Seedi, H.R., 2020. Recent insights into chemical and pharmacological studies of bee bread. Trends in Food Science & Technology, 97, pp.300-316.
  1. English editing is highly recommended.

Author Response

1. The storage conditions during five and ten days should be mentioned.

Answer: Caviar was packed in glass jars with 100g volume with twist off cap and was stored in a refrigerator at the temperature of 10 degrees C, in darkness. The experiments on antioxidants were performed after 5 and 10 days of storage.

2. Line 56: “Beebread supplementation is the most beneficial way to make a polyphenols-rich diet.” rephrase.

Answer: Regular consumption of beebread can be one of the methods for supplementing the daily diet with antioxidative compounds.

3. The acidity of the product increased with the storage time, is it favored for the consumers?

Answer: Acidity changes shown in the study did not result in reduced sensory assessment of caviar. Acidity changes did not result in delamination of the product, leakage of the content of the balls or changes in the taste or smell. After 10 days, the honey aroma and taste were most pronounced.

4.Line 5-59: “Only two publications had been undertaken to study its composition through chromatography coupled with mass spectrometry”. In fact, several published papers presented on this subject as the following:

Answer: This sentence is obviously erroneous and it shall be corrected as follows: Regular consumption of beebread can be one of the methods for supplementing the daily diet with antioxidative compounds. The bee bread composition varies according to the origin of the pollen but is mainly composed of water, proteins, carbohydrates, lipids, inorganic elements and various other minor components such as decanoic acid, gamma globulin, nucleic acids, vitamins B and C, pantothenic acid, biopterin, neopterin, acetylcholine, and reproductive hormones, among others [8*]. Considerable amount of information can be found in literature on the types of polyphenol compounds contained in bee bread [5,6,7]. The most important of those compounds are: p-coumaric acid (367 µg/g), kaempferol (492 µg/g), isoramnetin (1086 µg/g) among phenolic compounds, as well as ferulic acid, caffeic acid, apigenin, and quercetin present in trace amounts, were identified in the composition of beebread.

  • [5] Sobral, F., Calhelha, R.C., Barros, L., Dueñas, M., Tomás, A., Santos-Buelga, C., Vilas-Boas, M. and Ferreira, I.C., 2017. Flavonoid composition and antitumor activity of bee bread collected in northeast Portugal. Molecules22(2), p.248.
  • [8*]Giroud B., Vauchez A., Vulliet E., Wiest L., Buleté A. Trace level determination of pyrethroid and neonicotinoid insecticides in beebread using acetonitrile-based extraction followed by analysis with ultra-high-performance liquid chromatography-tandem mass spectrometry. Chromatogr. A. 2013;1316:53–61. doi: 10.1016/j.chroma.2013.09.088.
  • Hryniewicka, M., Karpinska, A., Kijewska, M., Turkowicz, M.J. and Karpinska, J., 2016. LC/MS/MS analysis of α‐tocopherol and coenzyme Q10 content in lyophilized royal jelly, beebread and drone homogenate. Journal of mass spectrometry51(11), pp.1023-1029.
  • Othman, Z.A., Wan Ghazali, W.S., Noordin, L. and Mohamed, M., 2020. Phenolic compounds and the anti-atherogenic effect of bee bread in high-fat diet-induced obese rats. Antioxidants9(1), p.33.

5. Using sodium alginate, and calcium chloride during process is safe?, explain

Answer: Alginate is widely used in various industries such as food, beverage, textile, printing, and pharmaceutical as a thickening agent, stabilizer, emulsifier, chelating agent, encapsulation, swelling, a suspending agent, or used to form gels, films, and membrane [46,47]. Sodium alginate is the most common salt of alginate [48].The U.S. Food and Drug Administration (FDA) classifies food grade sodium alginate as GRAS (generally regarded as safe) substance in Title 21 of the Code for Federal Regulations (CFR) and lists its usage as an emulsifier, stabilizer, thickener, and gelling agent. The European Commission (EC) listed alginic acid and its salts (E400–E404) as an authorized food additive [41]

[46] Senturk Parreidt, T.; Müller, K.; Schmid, M. Alginate-Based Edible Films and Coatings for Food Packaging Applications. Foods 2018, 7, 170.

[41] Younes, M.; Aggett, P.; Aguilar, F.; Crebelli, R.; Filipiˇc, M.; Jose Frutos, M.; Galtier, P.; Gott, D.; Gundert-Remy, U.; Georg Kuhnle, G.; et al. Re-evaluation of alginic acid and its sodium, potassium, ammonium and calcium salts (e 400–e 404) as food additives. EFSA J. 2017, 15, 5049.

[47] Kim, Y.J.; Yoon, K.J.; Ko, S.W. Preparation and properties of alginate superabsorbent filament fibers crosslinked with glutaraldehyde. J. Appl. Polym. Sci. 2000, 78, 1797–1804.

[48] Yoo, S.; Krochta, J.M. Whey protein–polysaccharide blended edible film formation and barrier, tensile, thermal and transparency properties. J. Sci. Food Agric. 2011, 91, 2628–2636.

6. Line 151 “also propolis was hydrated”, the relationship between propolis and caviar here is not clear, please clarify.

Answer: Bee bread is a mix of pollen, honey and bee saliva that has been fermented. Propolis is a different bee product and it should not be mentioned here. The study was conducted concomitantly on other bee products, and the conclusions were erroneously rewritten. Hydration primarily concerns flower pollen, and in addition, honey particles with proteins originating from bee saliva are dissolved in water after a few days, creating a uniform mass.

7. The plant origin of the bee bread should be included.

Answer: Beebread has been collected from the hives located near a buckwheat plantation. Hence, the majority of flower pollen observed under microscope originated from buckwheat flowers (Fagopyrum sagittatum). Therefore, the beebread was dissolved in buckwheat honey.

8. The content of phenolic compounds in beebread caviar increased with the storage time, explain.

Answer: With storage time, the amount of tested total phenol compounds increases. 2 causes for this state are expected: firstly, the process of dissolution and extraction of compounds contained in beebread occurs in the capsules, which may result in better dissolution and higher values of these compounds as converted into gallic acid after the period of 10 days. The second reason may be the process of honey fermentation, which may occur spontaneously inside the capsule. Honey fermentation results in increased acidity of the product and increased amount of polyphenol compounds.

9. Is the final product expected to be safe, useful, cheap, and attractive of consumers?

Answer: The use of edible coatings made of sodium alginate is a new trend that is increasingly commonly used in production of convenience foods and in environmentally friendly packaging. The present study utilized sodium alginate to produce caviar made of a mix of beebread and buckwheat honey. Beebread in this form is a more attractive product, as it can be used without earlier preparation (dissolved in water, juice, at the correct temperature and within the correct duration). Beebread in the form of caviar is also an attractive form of functional food with high content of natural bioactive compounds, which can be served as an appetizer or as an additive to cakes, salads and desserts.

10. Figure1: The conditions of each step should be included and the figure must be done with high resolutions.

11. References section’: styles should be done based on the journal requirements.

12. The recent interesting reference may help in updating information in the introduction.

  • Khalifa, S.A., Elashal, M., Kieliszek, M., Ghazala, N.E., Farag, M.A., Saeed, A., Xiao, J., Zou, X., Khatib, A., Göransson, U. and El-Seedi, H.R., 2020. Recent insights into chemical and pharmacological studies of bee bread. Trends in Food Science & Technology, 97, pp.300-316.

13. English editing is highly recommended.

 Answer 10-13: The sentence has been corrected according to reviewer's suggestions

Reviewer 2 Report

The ususal contetn og alginate used for imobilization in food preparation is 1%. The Authors found that for their particular product development higher amount is more appropriate (2%). The research was carried out successfully and the system alginate-beebread-honey was well elaborated trough sensors and physical/chemical analysis. In this sense the results are convincing.

The Authors should used first Author name to make a referencing. This should be checked trough the whole manuscript.

Line 171: …to the study by Socha et al.     instead of     …to the study by( 19)

Line 187. Socha et al. (19) also claimed….     Instead of   (19) also claimed….    

The research is in the line with the current trends in food industry and it the result could be observed as a new convenient and functional food product.

Author Response

The Authors should used first Author name to make a referencing. This should be checked through the whole manuscript.

Line 171: …to the study by Socha et al.     instead of     …to the study by( 19)

Line 187. Socha et al. (19) also claimed….     Instead of   (19) also claimed….   

Answer:  The remarks have been corrected according to the reviewer's suggestions

Reviewer 3 Report

The manuscript entitled "Preparation of beebread caviar from buckwheat honey through immobilization with sodium alginate” addressees some important questions regarding the preparation of caviar from beebread analyzing its storage stability and nutritional properties. The interpretation of some comments of the manuscript needs to be clarified. Some points are:

  • Abstract – line 27. The authors should indicate the standard compound used to measure the total phenols.
  • Abstract – line 28. The authors should determine the total phenolic content at different days until 5 days after preparation to demonstrate that “it remained stable until 5 days after preparation”.
  • Page 5 – lines 159-163. The paragraph is speculative and the interpretation of the results should be clarified. Why did the content of phenols increase after 5 days of storage? Some interpretations of the results should be included. The authors should include results about antioxidant activity at different days to demonstrate that “an increase of the content of phenols indicates a significant increase in the activity of bioactive compounds”. Sometimes a maximum plateau may be reached after a given concentration of phenols or the potential interactions may take place between the phenolic compounds and other components found on extracts, with either synergistic or inhibiting effects. Likewise, the analytical characterization of the content of phenols should be included. The composition of these natural antioxidants plays an important role in their activity.
  • Page 6 – line 175. The authors should check the format of the reference.
  • Page 6 – lines 185- 187. The paragraph should be reworded to improve the understanding of the reader.
  • Page 7 – line 220. It should be “…extract content decreased significantly…”
  • Page 9 – line 298. The immobilized beebread was stored at 7ºC and dark conditions? The authors should provide more details.
  • Page 9 - Analytical procedures – lines 299-316. The description of the experimental procedures employed should be included.

Author Response

  • Abstract – line 27. The authors should indicate the standard compound used to measure the total phenols.

Answer: Beebread caviar, obtained by immobilization on alginate carrier, contained 34,1 mg GAE/100g sample.

  • Abstract – line 28. The authors should determine the total phenolic content at different days until 5 days after preparation to demonstrate that “it remained stable until 5 days after preparation”.

Answer: In the study, TPC was measured after the product was prepared and after 5 and 10 days of storage. Polyphenols were not measured every 24h for 5 days.

  • Page 5 – lines 159-163. The paragraph is speculative and the interpretation of the results should be clarified. Why did the content of phenols increase after 5 days of storage? Some interpretations of the results should be included. The authors should include results about antioxidant activity at different days to demonstrate that “an increase of the content of phenols indicates a significant increase in the activity of bioactive compounds”. Sometimes a maximum plateau may be reached after a given concentration of phenols or the potential interactions may take place between the phenolic compounds and other components found on extracts, with either synergistic or inhibiting effects. Likewise, the analytical characterization of the content of phenols should be included. The composition of these natural antioxidants plays an important role in their activity.

Answer: Polyphenols content was tested following caviar production and during storage, but after 5 and 10 days. The experiment was done in three replications and in each case higher polyphenols content was observed after 5 days of storage. According to the authors this may be a result of processes occurring within the capsules. 2 causes for this state are expected: firstly, the process of dissolution and extraction of compounds contained in beebread occurs in the capsules, which may result in better dissolution and higher values of these compounds as converted into gallic acid after the period of 10 days. The second reason may be the process of honey fermentation, which may occur spontaneously inside a capsule. Honey fermentation results in increased acidity in the product and increased amount of polyphenol compounds.

  • Page 6 – line 175. The authors should check the format of the reference.

Answer: The sentence has been corrected according to reviewer's suggestions

  • Page 6 – lines 185- 187. The paragraph should be reworded to improve the understanding of the reader.

Answer: The sentence has been corrected according to reviewer's suggestions

  • Page 7 – line 220. It should be “…extract content decreased significantly…”

Answer: The sentence has been corrected according to reviewer's suggestions

  • Page 9 – line 298. The immobilized beebread was stored at 7 and dark conditions? The authors should provide more details.

Answer: Following the preparation, caviar was packed into glass jars with 100 g volume and twist off cap. Subsequently, it was stored in refrigeration conditions at temp. 7 ºC, in darkness, for 10 days. The jars were opened only once, on the day of testing.

  • Page 9 - Analytical procedures – lines 299-316. The description of the experimental procedures employed should be included.

Answer: Description of analytical methods has been extended as follows:

Proline determination was performed after its separation from other amino acids present in honey, with spectrophotometric method, utilizing a UV-VIS 1600 spectrophotometer (VWR International). The same spectrophotometer was utilized in cuvette tests of glucose, fructose and sucrose were obtained by the enzymatic determination method using a sucrose/d-glucose/d-fructose UV test no 716260 (Boehringer Mannheim, Germany).

Diastatic activity of honey was determined using spectrophotometry, in which insoluble starch conjugated with blue dye was used as the substrate. It is hydrolyzed by amylase, which leads to obtaining water-soluble fragments of starch chain, creating blue connections with the dye, and the absorbance of which is measured at wavelength of 620 nm. The solution absorbance is proportionate to the diastatic activity of the sample [9].

The beebread immobilized on alginate capsules was analyzed for the content of phenolic compounds by Folina–Ciocalteau method (AOAC, 1974). Briefly, 10 g of sample was mixed with 20 mL of methanol and the mixture was stirred for 30 min at 30 °C. Then 250 µL of supernatant, 250 µL of Folin–Ciocalteau reagent and 500 µL of 20% sodium carbonate in water were added in 4.00 mL of water. After 30 min, absorbance was measured at 760 nm using UV–vis spectrophotometer with methanol as the reference. Gallic acid (0–100 mg/L) was used to produce a standard calibration curve. The TPC was expressed in milligram of gallic acid equivalents (mg GAE/ml of extract).

Total acidity was determined according to Polish Standard [9], according to which 10 g of caviar was weighed, disintegrated and dissolved in 50 ml of distilled water, followed by 15 min shaking on a Vortex. Subsequently, the samples were titrated with 0.1M sodium hydroxide against phenolphthalein as an indicator to bright pink coloration. The titration was performed in three replications. Total acidity was calculated following a formula, including the sample size and the amount of sodium hydroxide used to produce discoloration.

The extract was determined using a HANNA HI 96803 digital refractometer (HANNA Instruments Poland). To make this possible, a sample was disintegrated to liquefy it and it was then placed using a plastic pipette on a dry and clean prism. The measurement result was automatically read and was performed at a calibration temperature of the device, that is 20℃. The measurement was done in three replications.

Sensory analysis was performed with five-point scale method [10], which includes five quality levels for each quality trait. Appropriate quality definition is assigned to each level. These definitions are specific for different types of products, thus they have to be strict and separate for the individual levels of the scale, so the assessing person could not have any doubts as to which level the product should be qualified. The point assessment was performed by a team of 5 people, and the results presented in the publication were averaged.

Round 2

Reviewer 1 Report

-

Reviewer 3 Report

  Thank you for your revised manuscript. Reviewer is pleased to see that authors have taken of the majority of the reviewers' comments. The manuscript would be publishable.